# In Silico Screening of Drugs That Target Different Forms of E Protein for Potential Treatment of COVID-19

**DOI:** 10.3390/ph16020296

**Published:** 2023-02-14

**Authors:** Gema Lizbeth Ramírez Salinas, Alejandro López Rincón, Jazmín García Machorro, José Correa Basurto, Marlet Martínez Archundia

**Affiliations:** 1Laboratorio de Diseño y Desarrollo de Nuevos Fármacos e Innovación Biotecnológica (Laboratory for the Design and Development of New Drugs and Biotechnological Innovation), Escuela Superior de Medicina, Instituto Politécnico Nacional, México City 11340, Mexico; 2Utrecht Institute for Pharmaceutical Sciences, Division Pharmacology, Utrecht University, 3553 Utrecht, The Netherlands; 3Julius Center for Health Sciences and Primary Care, Department of Data Science, University Medical Center Utrecht, 3553 Utrecht, The Netherlands; 4Laboratorio de Medicina de Conservación, Escuela Superior de Medicina, Instituto Politécnico Nacional, México City 11340, Mexico

**Keywords:** SARS-CoV-2, COVID-19, drug repositioning, in silico studies, E protein

## Abstract

Recently the E protein of SARS-CoV-2 has become a very important target in the potential treatment of COVID-19 since it is known to regulate different stages of the viral cycle. There is biochemical evidence that E protein exists in two forms, as monomer and homopentamer. An in silico screening analysis was carried out employing 5852 ligands (from Zinc databases), and performing an ADMET analysis, remaining a set of 2155 compounds. Furthermore, docking analysis was performed on specific sites and different forms of the E protein. From this study we could identify that the following ligands showed the highest binding affinity: nilotinib, dutasteride, irinotecan, saquinavir and alectinib. We carried out some molecular dynamics simulations and free energy MM–PBSA calculations of the protein–ligand complexes (with the mentioned ligands). Of worthy interest is that saquinavir, nilotinib and alectinib are also considered as a promising multitarget ligand because it seems to inhibit three targets, which play an important role in the viral cycle. On the other side, saquinavir was shown to be able to bind to E protein both in its monomeric as well as pentameric forms. Finally, further experimental assays are needed to probe our hypothesis derived from in silico studies.

## 1. Introduction

The coronavirus (CoV) belongs to the *Coronaviridae* family, subfamily *Coronavirinae*. They are RNA single-stranded and enveloped viruses. The subfamily *Coronavirinae* are divided in four genera: alpha, beta, gamma and delta coronavirus. The CoVs are able to infect different species of mammals, including human beings, mainly the beta coronavirus [1]. The first reports on endemic human CoV (HCoV) were published in the decade of 1960, when HCoV-OC43 and HCoV-229E were first described. Some years later, HCoV-NL63 and HCoV-HKU1 were discovered in 2004 and 2005 [2,3].

Some of the most known endemic CoVs in the last two decades include: (a) SARS-CoV that emerged in 2002 with about 8000 cases all over the world and with a mortality rate of 9.6%, (b) MERS-CoV that emerged in 2012 and infected about 2500 over the world and with a mortality rate of 40%, and (c) SARS-CoV-2 which shows a mortality of 6.9% [4], representing the lowest rate in comparison to other coronaviruses but with the highest efficiency of virus spread infecting 676,945,055 people and 6,777,045 of deaths (9 February 2023) [5], leading to an important social, economic and health problem all over the globe. The disease originated by SARS-CoV-2 has been denominated COVID-19. The pathology of COVID-19 has been characterized by intense, rapid stimulation of the innate immune response that triggers activation of diverse proteins such as inflammasome (NOD-like receptor family, pyrin domain containing 3 (NLRP3), NOD-like receptor family and among others) [6].

The CoV genome codifies four main structural proteins: spike protein (S), nucleocapsid (N), membrane protein (M) and envelope protein (E). All these proteins are necessary to assemble the virus and to release new viruses in the human cells [7,8]. These structural proteins are conserved from a subgenomic RNA. From all these structural proteins, E protein is the least abundant protein of the mature viral particles and is also the smallest [9,10]. Specifically, for the CoV, the E protein is particularly expressed during the cellular infection stage although only a small portion of this protein is incorporated into the virions [10,11].

The E protein of the CoV is an integral membrane protein which consists of 76 to 109 amino acids and has a molecular size within 8.4 to 12 kDa [12,13]. This protein shows a short N-terminal segment of 7 to 12 residues followed by a hydrophobic transmembrane domain (TMD) of 25 residues and ending up with a large hydrophilic C-terminal segment (segment 38–75) [13,14]. The E protein shows palindromic transmembrane helical hairpin around a pseudo-center of symmetry. The hairpin deforms lipid bilayers by way of increasing their curvature, while playing a fundamental role in viral budding [15]. Some other critical functions of E protein include replication cycle, including virion assembly, budding, release, and pathogenesis processes [16] The TMD manifests ion channel activity, while the C-terminal segment and N-terminal region participate in protein–protein interactions [17].

### 1.1. Homopentamer

This E protein is viroporin. These viroporins are known to form membrane pores [14,18,19]. It has been reported that during the viral infection, the viroporins oligomerize causing the disruption of the physiological homeostasis in the host cell while contributing the viral pathogenicity [20].

### 1.2. Ion Channel

The E protein of CoV is selective to cations which is related to its ionic channel properties, and it shows its preference for the cationic monovalent Na^+^ and K^+^. Moreover, E protein is essential and known for its role in the activation of the inflammatory NF-kB pathway [21]. The E protein also forms a calcium ion (Ca^2+^) channel in the endoplasmic reticulum Golgi apparatus intermediate compartment (ERGIC)/Golgi membranes [21]. The changes in calcium homeostasis in the intracellular environment leads to activation of the cytosolic innate immune signaling receptor NLRP3 inflammasome [6].

The activation of the inflammasome for the E protein of SARS-CoV was identified for the first time in porcine reproductive and respiratory syndrome virus (PRRSV). The blockage of the activity of the ionic channel with amantadine significantly inhibited the activation of the inflammasome which suggests the role of the E protein in the inflammation processes [13]. Additionally, it has been shown the role of the E protein in Ca^2+^ transportation in SARS-CoV, which triggers activation of the inflammasome [21]. Interestingly, it has been associated to the decrease of inflammatory cytokines in the absence of activity of ionic channel E in CoV as well as the inhibition or deficiency of the E protein [13,17,22].

Additionally, the biochemical evidence proposes that the capability of the E protein of the CoV to form homo-oligomeric conformers depends on its TMD [13,23]. In addition, the capacity of the E protein of CoV to assemble homopentameric structures is clearly important in the formation of a functional viroporin [13]. Recently, some point mutations in the TMD such as N15A and V25F have been found to eliminate the capability of ionic canalization of the viroporin; on the other hand, these mutations seem to block the oligomerization of the E protein in SARS-CoV [13,23]. The appearance of monomers in response to V25F clearly suggests that this residue plays a crucial role in oligomerization—in contrast to the N15A mutant, which reduces the pentamer formation [13,23]. The assembly of the homopentamer of E protein is crucial for virus replication [23].

### 1.3. N-Terminal

It has been determined that the N-terminal segment of the E protein is responsible for the activation of TLR2 in macrophages, and thus in the inflammatory signaling pathways [24,25,26]. For all these reasons, the N-terminal segment has been considered a potential target for inhibition. Recently, different experiments have been employed in order to identify proteins that are associated with the uncontrolled production of proinflammatory cytokines which could lead to a serious infection of COVID-19 [24].

### 1.4. Monomeric

The C-terminal region of E protein interacts with several other proteins:

(1) Interaction with PALS1:

The E protein by PDZ-Binding Motif (PBM) interacts with syntenin proteins which triggers the activation of the p38 MAPK and leads to the overexpression of inflammatory cytokines [27,28]. Additionally, the C-terminal domain of the monomeric E protein affects host intracellular activities through interference with the Golgi endoplasmic reticulum and intermediate compartment ER-Golgi [29]. The E protein monomer modulates intracellular activities of the host through the C-terminal domain (segment 38–75) [13]. In the C-terminal domain, there is the PDZ-domain binding motif (DLLV), which binds to the Protein Associated with Lin Seven 1 (PALS1) [30]. The interactions of the E protein of SARS and PALS1 protein introduced the relocation of PALS1 in the assembly site of the virus and interrupted the narrow binding sites to promote virus propagation [31]. The increased virulence of SARS-CoV-2 compared to SARS-CoV may rely on the increased affinity of its E protein for PALS1 [31,32]. The residues of E protein of SARS-CoV-2 that are involved in the formation of the complex E protein-PALS1 are: Val75, Leu74, Leu73, Asp72, Pro71 and Val70 [33].

(2) Interaction with BET:

Of worthy interest is that Gordon and collaborators found that the transmembrane E protein, which is likely resident on the endoplasmic reticulum-Golgi intermediate compartment and Golgi membranes, binds to bromodomain-containing protein 2 (BRD2) and bromodomain-containing protein 4 (BRD4) proteins, which are members of the bromodomain and extra-terminal (BET) domain family [34]. The C-terminal region of E protein mimics the N-terminal segment of histone H3, which is a known interacting partner of bromodomains [35].

On the other hand, it has been suggested that the interaction between BET with E protein can cause important changes in the genetic expression within the host cell, and thus, is also important for the viral cycle. BET proteins are also known to regulate immunity and inflammation mechanisms [36,37].

(3) Interaction with M protein:

The C-terminal segment of the E proteins is known to be important for the interaction with the C-terminal domain of the M protein, which is located at the cytoplasmic side of the endoplasmic reticulum-Golgi intermediate compartment, the budding compartment of the host cell [38]. The mentioned interactions are known to be considered important drivers for the envelope formation [39]. The interaction of the E protein with both PALS1 and BET proteins leads to proinflammatory activation mechanism, while interaction with M protein modulates viral budding processes and the release of newly formed viruses.

COVID-19 is characterized by an excessive production of proinflammatory cytokines, yielding in some cases to acute pulmonary damage which is highly associated with the mortality of the patients. Even though innate immunity cells produce multiple inflammatory cytokines during the SARS-CoV-2 infection, Karki and collaborators found that the combination of TNF-α and IFN-γ induces inflammatory cellular death [40]. The SARS-CoV-2 infection causes cardiac dysfunction induced by proinflammatory cytokines. Additionally, TNF has been also associated with cardiac dysfunction, while inducing systolic dysfunction [37].

Due to the strong evidence that the E protein is crucial in the modulation of diverse processes, in this work we propose diverse in silico approaches to study the binding of potential inhibitors on this protein. Moreover, here we propose to investigate different forms of the E protein, such as monomer as well as pentamer. We also considered different potential sites of inhibition (1) ionic channel (pentamer), (2) N-terminal (pentamer), (3) C-terminal (monomer).

It has been stated in the literature that the C-terminal region is a very flexible segment. Two conformations of the monomer have been denominated: the first one is where the C-terminal region shows the shape of a curve or harpin, and it is embedded in the membrane [41,42]; the second conformation of the C-terminal region is exposed to solvent [41]. In the present work, we have investigated both conformations as potential sites of inhibition. Due to the sanitary emergency of the COVID-19 pandemic, it became quite important to discover new potential drugs for the treatment of this disease, although it is known that drug discovery is a time-consuming and high-investment process [43]. Nowadays, drug repositioning represents an effective strategy to find new uses for existing and already probed drugs which makes it a highly efficient, low cost and riskless procedure [43]. In this work, we have considered an in silico strategy from a large database of compounds while using ADMET screening analysis in order to identify potential drugs that could inhibit different conformations of the E protein of SARS-CoV-2.

## 2. Results

### 2.1. Molecular Modeling of E Protein of SARS-CoV-2

Quaternary structures of the homopentamer and monomeric E protein of SARS-CoV-2 were modeled. Figure 1 shows the quaternary structure of this homopentamer and monomeric forms, and ionic channels, N-terminal and C-terminal. As stated in the Methodology (Section 4.1), the tri-dimensional model of the E protein of SARS-CoV2 was built by using the crystal structure of the E protein of SARS-CoV (PDB: 5 × 29) and employing Modeller 10.1 Software. Alignment of the query vs. template is shown in the Appendix A.

### 2.2. In Silico ADMET Analysis

We screened 5852 compounds from the Zinc database through ADMET analysis, discarding the compounds that did not fulfill the safety requirements (3697 ligands). Furthermore, docking analysis was performed with the remaining 2155 compounds.

### 2.3. Molecular Docking Analysis on the Different Targets

Different docking analyses were carried out, considering different conditions: (a) homopentameric form as an ion channel itself and the N-terminal segment of this form, (b) monomer form which includes two binding sites: C-terminal solvent-exposed and hairpin. Four molecular docking studies were carried out, which are described (Appendix A).

#### 2.3.1. Molecular Docking Analysis on E Protein Form Homopentamer (Ionic Channel Region)

As can be seen in the Figure 2A, lumacaflor binds to E protein through Pi-Pi interactions which include the following residues: Phe26E, Phe26A y Phe23E, alkyl interactions with the residues Val29A, Val25A, Ala22E, Leu19E, Ala22C and Ala22B, halogen interactions with the residue Ala22D, and van der Waal’s forces with the following residues: Phe23C, Phe26D, Phe26C, Phe26B, Val25D, Leu19C and Ala22A. Figure 2B shows that saquinavir binds to E protein through van der Waal’s interactions by interacting with the residues Phe26B, Ala22B, Leu19A, Phe26C, Phe26D, Phe23C, Ala22D, Leu18D, Leu19C, Phe26A, Leu18E, Leu19D, Leu18B, Leu19E, Val25A, Phe23E, Leu18C, Asn15B, Leu21C and Pi-Sigma interaction with Ala22C, alkyl interactions with Ala 22A, Leu18A, Ala22E, Leu19B, and finally Pi-Pi interaction with Phe26E. In the Figure 2C is depicted that nilotinib interacts with E protein through diverse interactions such as: (a)van der Waal’s interactions with the following residues: Leu28D, Leu27C, Val25D, Leu19A, Phe23A, Phe26B, Phe26A, Phe26D, Thr30C, Phe26E, Val29C, (b) Pi-sigma interactions with the amino acids: Ala22B and Ala22D, (c) Pi-Pi interactions with Phe26C, (d) Alkyl interactions with Ala22C, Leu19C, Ala22A and Val29D, and (e) Hydrogen bonds with Phe23C and Phe26C. The zafirlukast (Figure 2D) binds to E protein through Pi-Pi interactions which include the following residues: Phe26C, Phe26A, Phe26E and Phe23E, the amino acid Val29C forming Pi-sigma interactions, the van der Waal’s interactions with Thr30C, Thr30B, Thr30E, Phe23C, Ile33C and Val29E, and these amino acids Val29D, Val29A, Val25D, Phe26B and Phe26D forming alkyl interactions. 

#### 2.3.2. Molecular Docking Studies on E Protein Form Homopentamer (N-Terminal Region)

The amino acids of the N-terminal region of E protein in the homopentamer form interact with alectinib (Figure 3) while showing the following interactions: Asn15D (Hydrogen bond), Glu8A (Salt Bridge), Thr11A, Glu7B (Pi-Sigma interactions), Val5B, Leu12A, (alkyl interactions), Thr11B (carbon hydrogen bond), Asn15E, Val14E, Thr11D, Asn15, Glu7B, Gly10B, Ser6B, Glu7A and Ser3B (van der Waal’s forces).

#### 2.3.3. Molecular Docking Analysis on E Protein Form Monomer (C-Terminal Solvent Exposed)

Figure 4A shows the molecular interactions between the E protein monomeric (with the solvent-exposed C-terminal region) and the compound irinotecan. The drug interacts with the E protein through van der Waal’s interactions with the amino acids Leu51, Lys53, Phe56, Arg69, Ser67, Ser68 and Val70. while the amino acids Ser60 and Asn64 form hydrogen bonds, and finally Pi interactions are observed between the amino acids Tyr57, Lys63, Val75 and Asp72. The molecular interactions that form in the saquinavir-E-monomer complex are as follows (Figure 4B): Hydrogen bond (Ser60, Asn64 and Arg69), van der Waal’s force interactions (Leu51, Lys53, Phe56, Tyr59, Lys63, Ser67, Ser68, Val70, Asp72 and Val75), Pi-alkyl interactions (Leu74) and Pi-Pi T-Shaped interactions (Tyr57). The amino acids (Figure 4C) that interact to form the dutasteride-E-monomer complex are the following: Phe56, Tyr59, Ser60, Lys63, Asn64, Ser67, Ser68, Leu74 (van der Waal’s forces), Arg69 (hydrogen bond and Pi interaction), Val70 (Pi-alkyl Interaction), Pro71, Asp72 and Val75 (halogen interaction).

#### 2.3.4. Molecular Docking on E Protein Form Monomer (Harpin)

The drugs irinotecan, nilotinib, and saquinavir, bind to the E monomer in the hairpin conformations showing affinities of −8.2 kcal/mol, −8.2 kcal/mol and −7.7 kcal/mol respectively. Figure 5A shows the molecular interactions between irinotecan and the hairpin. Four main types of interactions are formed include: Pi-alkyl interactions (Ile46, Val49, and Leu51), hydrogen bonds (Ser55, Tyr57, and Tyr59), a carbon hydrogen bond (Val47) and van der Waal’s (Phe20, Phe23, Leu27, Val58, Leu73 and Leu74). Nilotinib (Figure 5B) interacts with the following amino acids of the E protein: Leu51, Val58, Val62, Arg69, Val70, Pro71 (Pi-alkyl interaction), Ser55 (van der Waal’s forces), Tyr57, Tyr59, Leu74, (hydrogen bonds) and Val75 (hydrogen bond and Pi-Sigma interaction). Figure 5C shows the interactions between saquinavir and the E monomer, the interactions that mediate this binding are hydrogen bonds (Ser55, Tyr57, Tyr59 and Val75), van der Waals forces (Val52, Phe56, Val58 and Pro71), Pi-alkyl interaction (Leu51, Arg61, Val70, and Leu74).

After analyzing the molecular docking results, some compounds were selected based on the affinity criteria and considering critical residues at the site of interest. Table 1 shows the selected compounds with the respective affinity calculated from the dockings; the affinities are shown in bold. The selected complexes were selected to perform MD simulations.

Pharmacokinetic properties of the compounds mentioned in Appendix A.

### 2.4. Molecular Dynamics Simulations

Sixteen molecular dynamics (protein–ligand complexes) of 150 ns in total were performed, in which six are MD simulations of the homopentamer with compounds targeting the ion channel, which were named with the following nomenclature: lumacaftor (E-IC_lumacaftor), nilotinib (E-IC_nilotinib), dutasteride (E-IC_dutasteride), naldemedine (E-IC_naldemedine), zafirlukast (E-IC_zafirlukast) and saquinavir (E-IC_saquinavir), a MD simulation of the homopentamer without ligand (E-IC-Nt). Alectinib was selected for MD simulation targeting the N-terminal site of the E protein homopentamer (E-Nt_alectinib). For the case of dockings directed to the C-terminal site, simulations were explored in the two conformations. For the solvent-exposed C-terminal conformation, four MD simulations were performed: saquinavir (E-CtS_saquinavir), irinotecan (E-CtS_irinotecan), dutasteride (E-CtS_ dutasteride), and the MD simulation without ligand (E-CtS). Finally, the dynamics performed for the hairpin include: irinotecan (E-CtH_irinotecan), nilotinib (E-CtH_nilotinib), saquinavir (E-CtH_saquinavir) and the MD simulations without ligand (E-CtH). The results of the different MD simulations are described below. Figure 6A shows the RMSD results from the trajectories, from which it could be observed that simulations fluctuate in the range of 6 to 8 Å along the 150 ns MD simulations of the E protein (homopentamer form ion channel) and N-terminal site. RMSD results from the trajectories (Figure 6B), from which it could be observed that simulations fluctuate in the range of 8 to 11 Å along the 150 ns MD simulations of the E-monomer solvent-exposed C-terminal whereas in MD simulations where the ligands were directed to the hairpin show values range from 6 to 9 Å (Figure 6C). As we could observe from the RMSD results, the monomeric forms show higher values since they are known to be more flexible, whereas the homopentameric assembly form tends to be more stable. This observation is in agreement with previous studies published by Kuzmin and collaborators [41].

### 2.5. Binding Free Energy Calculation Using MMPBSA Approach

The binding energy of E protein and the analyzed compounds were calculated. From these results, we could observe that saquinavir, nilotinib, dutasteride and alectinib bind with high affinity to the homopentamer form (Table 2).

In addition, dutasteride, irinotecan and saquinavir also bind to the monomer with the highest Gibbs free energy value in the C-terminal region (Table 3). Molecular interactions have been analyzed for the compounds marked in bold (Table 2 and Table 3).

Diverse molecular interactions were tracked along the different MD simulations.

#### 2.5.1. Monomeric Form

In the E-CtS_dutasteride complex we can observe different molecular interactions which include: Pi-alkyl interactions between residue Tyr59 and the ligand, additionally, this residue interacts with different rings of the ligand and the atoms C39 and C19. Furthermore, the residue Tyr59 shows a Pi-Sigma interaction with the atom HC, which is conserved in the MD simulation. Another alkyl interaction is formed between the atom CB of Val70 and the atom C39 of dutasteride, although this interaction is unstable. The interaction between irinotecan and the hairpin is towards one saline bridge formed between Val 75 (OT1) and H15 of the irinotecan; such an interaction is mostly conserved along the trajectory. On the other hand, in the binding of the ligand irinotecan on the hairpin, we could detect some important interactions which include: Leu74 (VdW), Leu73 (CHB) and Val75 (SB).

For the case of the complex E-CtH_saquinavir we could identify some hydrogen bonds that are stable along the trajectory, one of which is formed between Val75 (OT1) and the H89 of the drug. In addition, a hydrogen bond interaction was formed between the atom HG1 of Ser55 and the atom O43 of the ligand; such interaction is observed in almost half of the simulation. Another hydrogen bond that is formed between the atom HH of Tyr59 (E protein) and the atom O43 of saquinavir which is conserved half of the of the trajectory and it showed some other VdW interactions with the residues: Ile46, Tyr42 and Val58 Additionally, we could observe a saline bridge between the saquinavir and Val75 which is conserved mostly along the trajectory.

#### 2.5.2. Homopentamer Form

In the complex E-IC_dutasteride it could be observed an alkyl interaction between Val17A (CB) of the pentameric form of the E protein and the atom C32 of the ligand dutasteride, which is mostly conserved along the trajectory. Additionally, other alkyl interaction is formed between Val12A (CB) and the atom C32 of the ligand, which is also conserved along the trajectory. Regarding the interactions between the compound alectinib and the homopentaric form of E protein, in which we could observe a saline bridge between the residue Glu8A, Pi-cation interaction with the amino acid Phe4B , and hydrogen bonds with the residues: Asn15D, Asn15E, and Asn15A. In the case of the binding of saquinavir on the homopentameric form it is mainly through van der Waal’s interactions with the following residues: Asn15B, Asn15D, and Phe26C. All the mentioned interactions are shown constantly along the trajectories. Regarding the interactions of nilotinib with E homopentameric, it shows constant interactions with these residues: Phe26B (Pi-Pi stacking interaction), Phe26D (Pi-alkyl interaction), Phe26E (Pi-alkyl interaction) and Phe26A (van der Waals).

## 3. Discussion

E protein sequences from SARS CoV (NP_828854.1) and SARS CoV-2 (BCA87363.1) have an identity of 94.7% and sequence similarity of 97.4% [8]. Importantly, E protein is crucial in the viral replication cycle [44] such as assembly [45], virion release [45,46], and viral pathogenesis, [47,48] induction of membrane curvature, inflammation and even autophagy [49]. Additionally, the possibility of finding E protein in the Golgi and endoplasmic reticulum compartments has been described; where it can interact with bromodomain proteins such as BRD2 and BRD4 (proteins that bind to acetylated histones to regulate gene transcription) [50]. The C-terminal E protein binds to the PDZ domain, which induces immune-pathological reactions and causes overexpression of inflammatory cytokines. Indeed, E protein plays an important role in the release of inflammatory cytokines, which causes the acute respiratory syndrome, and it is considered the main cause of the death of patients with COVID-19 [51,52]. Thus, this protein can be considered as an interesting pharmacological target for potential antiviral drugs [16,17]. It has been also described that both the homopentamer and monomer forms play a role in the viral replication, as well as in the activation of pro-inflammatory signaling pathways, which in many cases are involved in the exacerbation of COVID-19. For all these mentioned ligands, targeting the C and N terminal of the ion channel has been considered [24,32,37].

From the binding energy simulations (MM–PBSA approach), we could identify the best evaluated compounds on the different forms of E protein (Table 2 and Table 3).

Interactions between the monomeric and homopentameric forms are described below:

### 3.1. Monomeric

Considering the free binding energy simulations (MM–PBSA approach), we found that the compound dutasteride showed the highest energy on the C-terminal exposed to solvent; and for the hairpin, the most promissory compounds include irinotecan (−9.8740 kcal/mol) and saquinavir (−11.9547 kcal/mol). On the other side, considering the monomeric form, the compound irinotecan interacts with the residues: Leu73, Leu74 and Val75, and for the compound saquinavir, it shows interactions with Val75 and Leu74. The mentioned residues belong to the PBM of the E protein, which is known to be essential for the binding to the PDZ domain of the PALS1 protein [32,33]. The capacity of E protein to bind to the PDZ-binding domain has been associated with the virulence of the virus [13,41].

On the other hand, there is valuable information (protein interaction maps) about the involvement of C-terminal helical bundle as binding epitope for ligand binding, thus, corroborating our hypothesis regarding the potential role of a C-terminal helical bundle in the mediation of viral replication processes [52,53]. Interestingly the C-terminal region of E (can interact with bromodomains) is highly conserved in SARS and bat coronaviruses, which suggests that it has a conserved function [31,54]. This means that the compounds that interact in this region could have an effect on various coronaviruses.

### 3.2. Pentameric

We have performed diverse in silico studies which include docking analysis, MD simulations and free binding energy calculations (MM–PBSA) from which could depict the energy values of the following ligands: saquinavir −24.9147 kcal/mol, nilotinib −23.0781 kcal/mol, dutasteride −16.9153 kcal/mol and alectinib −116.67 kcal/mol showed the highest binding affinity to the homopentamer. Of worthy interest is that we could depict that alectinib showed the highest binding free energy (ΔG (−116.67 kcal/mol) from the molecular dynamics simulation studies in comparison to the other compounds. From the binding interactions, it could be observed that the mentioned ligand interacts with the residue Asn15 through hydrogen bond interactions, additionally it interacts with the segment of amino acid residues 7 to 12, which make it a very interesting target to inhibit the ionic channel and N-terminal at the same time. Additionally, it shows interactions with other residues which include: Glu8A and Phe4B.

Saquinavir seems to be an interesting ligand because it targets the spike protein and also 3C-like protease, as has been described previously [55,56]. Of worthy interest is that this ligand also binds to the E protein, and thus, it can be considered a multitarget ligand. Moreover, saquinavir is able to bind to both monomeric and homopentamer forms. Sarkar and collaborators reported that the homopentameric form interacts with the residues: Phe26 and Asn15 [57]. This fact would provide advantages over other compounds that only interact with one form (monomer or pentamer), since E protein could have conformational changes in the different steps of the viral replication cycle.

It has been reported that nilotinib appears to bind to the receptor binding domain RBD of the SARS-CoV-2 spike protein, and some in silico studies have been carried out to depict the interaction between tyrosine kinase inhibitors and SARS-CoV-2 protein. On the other side alectinib has been prescribed along antiviral treatment however clinical trials are needed in terms of treatment of cancer and treatment of COVID-19. Furthermore, it has been identified that the residue Phe26 is located in a key position better known as “bottleneck region” which is important for the function of the ion channel of the E protein [12,57]. On the other side, residues such as: Phe4, Glu8, Asn15, and Val25 which are known to be important for the function of E protein, Phe4 functions as a gate towards the ion channel and it has been suggested that residues Glu8 and Asn15 regulate the opening/closing of the channel [46,57,58]. Particularly, Val25 and Phe26 have been suggested as key residues in the binding for effective inhibitors for E protein [59]. In a general manner, nilotinib, saquinavir and alectinib are considered the most promising compounds that could effectively inhibit the homopentamer of the E protein. The blockage of ionic channels of E protein could significantly reduce viral pathogenicity [46].

Moreover, the primary sequence of E protein is quite conserved among the different coronaviruses, making it an interesting therapeutic target [32,34]. Despite the genome of SARS-CoV-2 having evolved constantly, while generating novel variants, E protein still shows a high global conservation grade of 99.98% in which mutations are extremely infrequent, and is present in less than 0.3% of total sequences [54] in comparison to other structural proteins, such as spike protein which shows 2671 changes in 1132 of the 1272 spike amino acids [54].

Nowadays, targeted drug repurposing represents a very useful strategy to identify libraries of pre-existing molecules or approved drugs that could prevent COVID-19 [60,61].

Of worthy interest is that the ligands we proposed in this work could also bind to different SARS-CoV-2 variants as well as another coronaviruses, meaning they can be considered as potential drugs for COVID-19 treatment.

## 4. Methods

### 4.1. Molecular Modeling

Based on the primary sequence of the E protein of SARS-CoV-2, it was possible to build the transmembrane region (TMD) of the E protein of this virus, by employing the crystal structure of E protein from SARS-CoV PDB: 5X29 as template [62]. The identity percentage between the sequence P0DTC4 [63] (sequence of the E protein of SARS-CoV-2), and the crystal structure is 89.0%. Alignment of the query vs. template is shown in the Appendix A. The three-dimensional homology model of the homopentamer of the E protein of SARS-CoV-2 was built by using Modeller 10.1 Software [64]. The monomeric form of E protein was also modeled, for this purpose Modeler 10.1 Software [64] was employed, using the same template. Additionally, the scripts model-multichain-sym.py and model-loop.py were employed to obtain and refine the structure, yielding different 3D models which were built by means of the Modeller 10.1 program. Each of the models was first optimized with variable target function method (VTFM) with conjugate gradients (CG), and then, they were refined using molecular dynamics (MD) and simulated annealing (SA), while employing a slow refinement process of about 300 cycles during the whole optimization. CHARMM-22 parameters were employed to reproduce the protein geometry in the Modeller environment. Finally, the best evaluated model was selected using the discrete optimized protein energy (DOPE) method (GA341) [64,65,66].

### 4.2. Database Search and In Silico ADMET Analysis

We downloaded 5852 molecules from the Zinc database [67]. The molecules were obtained from the following subsets: FDA drugs (1604 FDA drugs, per Drug Bank), and world-not-FDA (4248 Other Drugs approved but not by the FDA).

In the process of virtual screening, software such as: DataWarrior program V5.5.0 [68] and SwissADME free web tool [69] were used to predict different drug-likeness parameters such as: physicochemical properties, solubility, and pharmacokinetics (absorption, distribution, metabolism, excretion and toxicity ADMET). From this analysis of 5852 drugs after some were discarded, the remaining 2155 compounds had to fulfill the following properties: high or medium lipophilicity(LogP > 1), bioavailability score (>0.11), Pains # Alert (0), Brenk # alert (≤2), Lipinski # Violation (≤2) and synthetic accessibility (<6.5) [70,71,72]. The filtered database is attached in Appendix A.

### 4.3. Docking Analysis

Molecular docking studies of the 2155 ligands were carried out using Autodock Vina (version 1.2) program [73]. The docking procedure was validated by comparing the molecular interactions of amantadine on SARS-CoV-2 reported previously [74]. For all the docking simulations, the E protein was rigid and the ligands flexible. The drugs with the highest affinity and those that reach key amino acids that influence the E protein function were selected. Docking analyses were carried out on homopentameric and monomeric forms of E protein. For the case of homopentameric E protein, dockings were focused to the ionic channel (active site is formed by the following residues: Glu8, Thr11, Leu12, Val14, Asn15, Val17, Leu18, Leu19, Phe20, Leu21, Ala22, Phe23, Val24, Val25, Phe26, Leu27, Leu28, Val29, Thr30, Leu31, Ala32, Ile33, Leu34, Thr35, Ala36, Leu37, Arg38, Leu39, Ala40, Tyr42, Ala43, Ala44, Ile46, Val47, Val49, Leu51, Pro54, Val56, Tyr57, Ser60, Arg61, Lys63, Asn64 and Leu65) as well as the N-terminal region (residues 7 to 12). For both conformations of the monomeric form, dockings were focused to the C-terminal segment (residues 70 to 75). The docking process on the homopentamer (ionic channel) was carried out considering the following parameters: grid box (25.0 Å × 30.0 Å × 30.0 Å) centered at (40.0, 63.0, 55.0) Å, and for the N-terminal: grid box (20.0 Å × 40.0 Å × 20.0 Å) centered at (20.0, 63.0, 55.0) Å. For the monomeric C-terminal harpin, grid box (25 Å × 25 Å × 25 Å) centered at (−20.0, 5.0, −1.0) Å, and finally for the C terminal region conformation (exposed to the solvent), a grid box (25 Å × 25 Å × 25 Å) centered at (−20.0, 5.0, −1.0) Å. For all the procedures, an implicit solvent function was employed, as well as the following parameters: num_modes = 100, energy_range = 6 and exhaustiveness = 25, and Monte Carlo force field. At the end of this procedure, the conformation that showed the highest Gibbs free energy was selected for further studies [75].

### 4.4. Molecular Docking Studies and Visualization of the Results

The process of selection was carried out by using a Perl script to obtain the binding affinity of each of 2155 compounds in just one file. Once docking simulations were finished, Discovery Studio Visualizer [76] and Chimera [77] programs were used to visualize the binding site of these ligands and their molecular interactions. Afterwards, ligands that showed the highest affinity (more negative) were selected. Additionally, surrounding residues could be detected from these theoretical studies.

### 4.5. Molecular Dynamics Simulations

Three E protein MD simulations (without a ligand) and sixteen MD simulations of E protein–ligand complexes were carried out by considering the results from docking analysis that showed the highest affinities and showed interactions with key residues. All these complexes were prepared by using Charmm-GUI Software [78,79] and embedded in a POPC membrane (1-palmitoyl-2-oleoyl-sn- glycero-3-phosphocholine) [80] which is native for this type of cell membrane. MD simulations were carried out by means of the NAMD program and using the known inputs for NAMD and standard scripts for MD simulations [81].

The method of particle-mesh Ewald (PME) was used for the calculation of the electrostatic potential energy. A no-bonded cutoff of 12 Å, switchdist of 10 Å and pairlistdist of 16 Å were implemented for these long-range interactions. MD simulations were solvated (TIP3 model) and neutralized up to a final concentration of 0.15 M NaCl in the equilibration step. The equilibration protocol consisted in six minimization steps, reaching a total of 2.5 ns of equilibration time; an NTV (constant volume and temperature) protocol was applied. For the production step an NTP (constant temperature and pressure) ensemble was maintained with a Langevin thermostat (310 K) and anisotropic Langevin barostat (1 atm). For these last two steps, an integration time step of 2 femtoseconds (fs) was used, with all the bond lengths involved, and used the CHARMM36 force field [78]. Finally, MD simulations were run 150 ns.

### 4.6. Structural Analysis of MD Simulations

Structural analysis of the E proteins (from the MD simulations) was carried out by employing the Carma program and considering the alpha atoms of the structure [80]. These structural analyses include calculation of the following parameters: RMSD (root mean square deviation) [82]. Additionally, we performed a structural comparison for each of the forms of the E protein: homopentamer, hairpin, C-terminal exposed to solvent (monomeric). For all the cases, the protein (without a ligand) conformation was used as a reference.

### 4.7. Binding Free Energy Calculations of the E Protein-Ligand Complexes

E protein–ligand complexes binding free energy (ΔG_bind_) were estimated for the 100 ns MD simulation trajectories. A stride of 10 was considered for the calculations, resulting in about 100 frames for the ΔG_bind_ analysis. The estimations were carried out employing molecular mechanics combined with the Poisson–Boltzmann surface area (MM–PBSA) method. MM–PBSA was applied by the Calculation of Free Energy (CaFE) plugin [83] implemented to the VMD program [84,85,86]. For the MM–PBSA calculations, we have considered the most stable part of the trajectories, for each of the cases.

## 5. Conclusions

In this work we have employed in silico methods (docking analysis, molecular dynamics simulations and MM–PBSA calculations) to identify compounds with potential E protein interaction from SARS-CoV-2. From these studies, we could identify five compounds: irinotecan, alectinib, saquinavir, nilotinib and dutasteride which were best evaluated. Particularly, alectinib could inhibit the functions of the ion channel as well as avoiding the binding of other proteins involved in pro-inflammatory processes that could trigger the cytokines cascade, which in consequence could lead to a serious and mortal illness. On the other side, saquinavir, nilotinib and alectinib were also considered as a promising multitarget ligand because they seem to inhibit three targets, which play an important role in the viral cycle. Additionally, saquinavir was shown to be able to bind to E protein both in its monomeric as well as pentameric forms so it could act in different steps of the viral replication cycle. Finally, further experimental assays are needed to probe our hypothesis derived from in silico studies.

## Consent for Publication

We give our consent to publish this manuscript.

## Figures and Tables

**Figure 1 pharmaceuticals-16-00296-f001:**
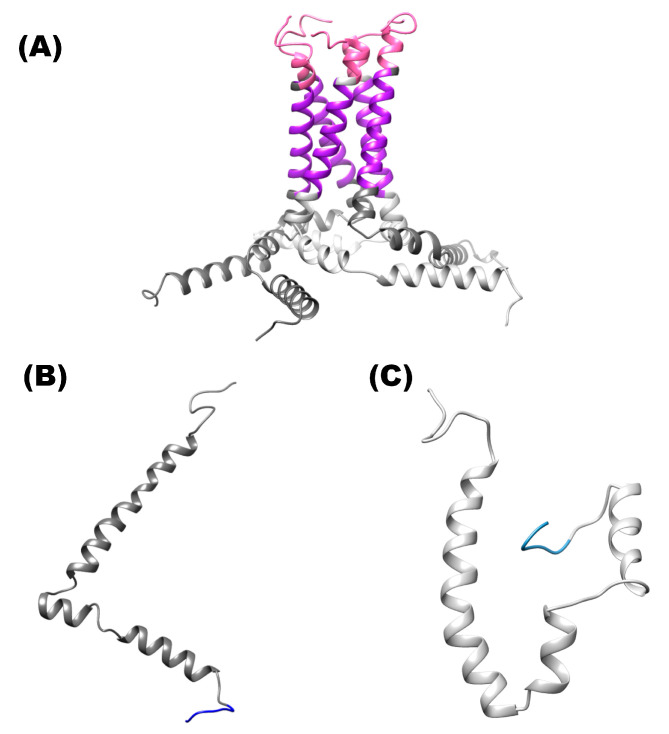
Structure of the E protein of SARS-CoV-2. (**A**) Shows the structure of homopentamer. (**B**) Three-dimensional structure of the protein is shown in ribbons and surface of the monomer C-terminal solvent-exposed. (**C**) Shows the ribbons of the monomer conformation (hairpin). Color pink (N-terminal site), color purple (ion channel) and color blue (C-terminal site).

**Figure 2 pharmaceuticals-16-00296-f002:**
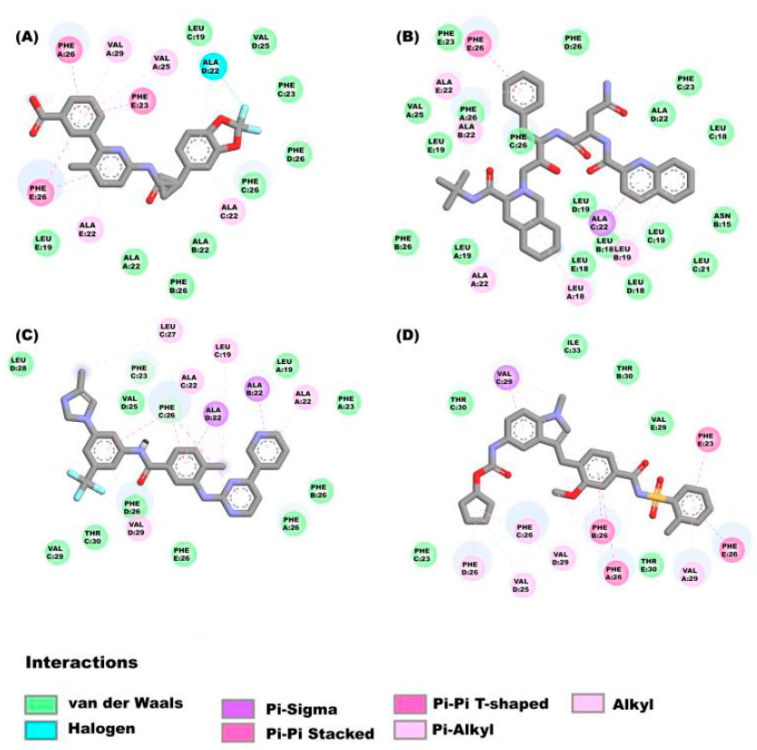
Molecular interactions between some ligands and the E protein from SARS-CoV-2: The molecular interactions of the compounds (**A**) lumacaflor, (**B**) saquinavir, (**C**) nilotinib and (**D**) zafirlukast are observed.

**Figure 3 pharmaceuticals-16-00296-f003:**
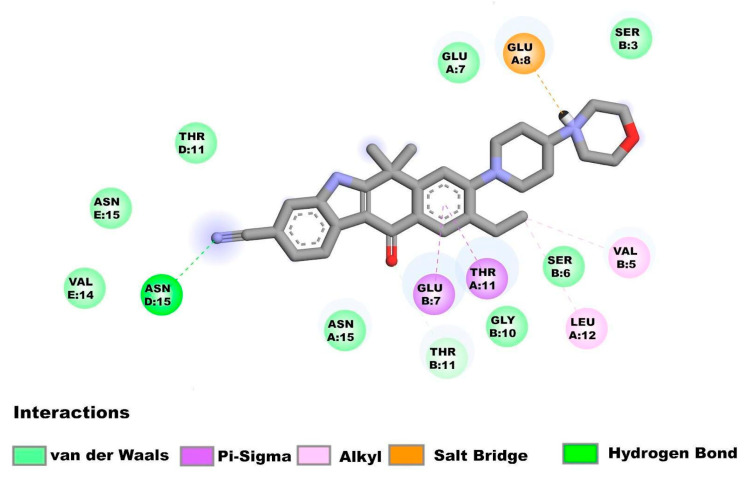
Molecular interactions between alectinib and the E homopentamer (N-terminal) from SARS-CoV-2: Molecular interactions of alectinib are shown.

**Figure 4 pharmaceuticals-16-00296-f004:**
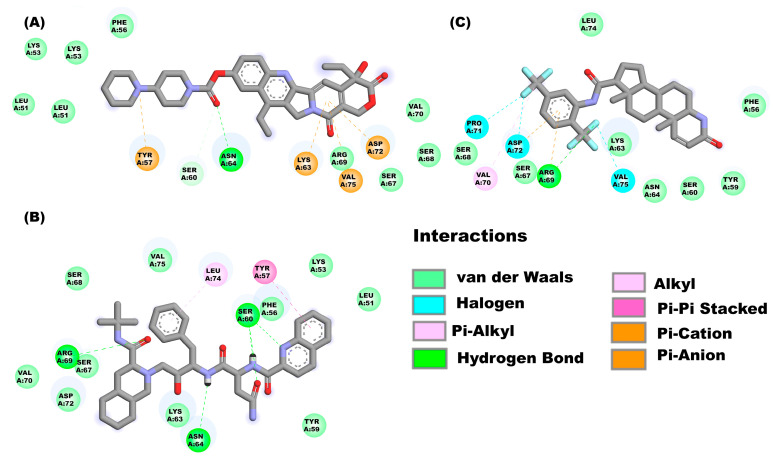
Molecular interactions between some drugs from the FDA subgroup and the E monomer from SARS-CoV-2: Molecular interactions of compounds (**A**) irinotecan, (**B**) saquinavir, and (**C**) dutasteride are observed.

**Figure 5 pharmaceuticals-16-00296-f005:**
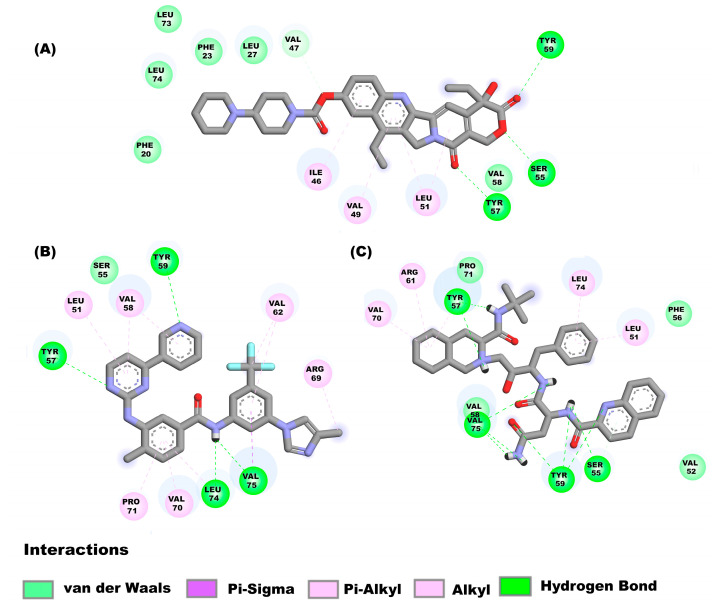
Molecular interactions between some drugs and the E monomer from SARS-CoV-2: Molecular interactions of compounds (**A**) irinotecan, (**B**) nilotinib, (**C**) saquinavir are shown.

**Figure 6 pharmaceuticals-16-00296-f006:**
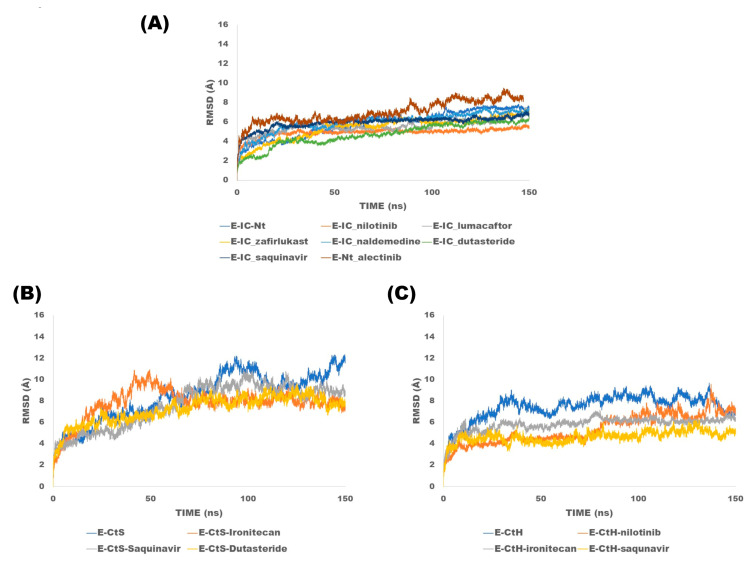
Root mean square deviations analysis of trajectories of the E protein simulations: (**A**) Ion channel and N-terminal, (**B**) C-terminal solvent-exposed and (**C**) hairpin: This figure shows the RMSD curves of each of the complexes.

**Table 1 pharmaceuticals-16-00296-t001:** Binding affinities of some compounds and considering different forms (homopentameric and monomeric). Results from docking analysis are reported in the Table below. ***** lumacaftor and nilotinib were bound at the solvent-exposed C-terminal site although no critical residues were identified in the binding sites. Thus, for that particular site (monomeric structure), these compounds were not considered for molecular dynamics simulations studies.

Binding Affinities of Some Compounds and Considering Different Forms				
	Homopentamer		Monomer	
Binding Site				
Drugs	Ion Channel	N-terminal	C-terminal solvent-exposed	C-terminal Harpin
	ΔG (kcal/mol)			
lumacaftor	−10.9	−9.1	−7.6 *	−7.6
nilotinib	−10.8	−9.8	−7.5 *	−8.2
dutasteride	−10.6	−8.9	−7.7	−8.1
naldemedine	−10.5	−8.7	−7.4	−7.7
zafirlukast	−10.2	−9.1	−7.0	−7.3
irinotecan	−10.1	−9.0	−8.0	−8.2
saquinavir	−9.9	−9.3	−7.9	−7.7
alectinib	−9.4	−9.6	−7.0	−7.0

**Table 2 pharmaceuticals-16-00296-t002:** Binding Free energy MMPBSA for the E-Homopentamer complexes: This table shows free binding energies of complexes between E protein (pentameric form) and ligand. Highest energies are shown in bold.

Binding Free Energy MMPBSA for the E-Homopentamer				
**Site Bind: Ion Channel**				
**Drugs**	**Complex Total (kcal/mol)**	**Receptor** **(kcal/mol)**	**Ligand** **(kcal/mol)**	** ΔG ** **(kcal/mol)**
Lumacaftor	−7885.9200	−7178.3400	−125.6700	−13.1500
Nilotinib	−7965.2763	−7537.2451	−404.9531	**−23.0781**
Dutasteride	−7392.0455	−7351.4972	−23.6330	**−16.9153**
Naldemedine	−7364.0697	−7341.7927	−15.1966	−7.0804
Zafirlukast	−7603.0986	−7358.9720	−241.1082	−3.0184
Saquinavir	−7582.0157	−7511.3579	−45.7430	**−24.9147**
**Site bind: N-terminal**				
Alectinib	−4203.9200	−4307.8700	−65.5600	**−116.6700**

**Table 3 pharmaceuticals-16-00296-t003:** Binding Free energy MMPBSA for the E-monomer complexes: This table shows free binding energies of complexes between E protein (monomeric form) and ligand. Highest energies are shown in bold.

Binding Free Energy MMPBSA for the E-Monomer				
**Binding Site: Solvent Exposed C-Terminal**				
**Drugs**	**Complex Total** **(kcal/mol)**	**Receptor** **(kcal/mol)**	**Ligand** **(kcal/mol)**	** ΔG ** **(kcal/mol)**
Dutasteride	−2175.4155	−2058.9008	−105.8080	**−10.7068**
Irinotecan	−2111.4271	−2062.2341	−44.8761	−4.3169
Saquinavir	−2206.3300	−2245.6700	−50.1300	−2.1400
**Binding site: Harpin**				
Nilotinib	−2238.1565	−2071.3184	−163.2539	−3.5842
Irinotecan	−2114.7274	−2059.0421	−45.8114	**−9.8740**
Saquinavir	−2122.5089	−2073.9448	−36.6094	**−11.9547**

## Data Availability

Data is contained within the article and Appendix A.

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
