# Peer review of "In Silico Screening of Drugs That Target Different Forms of E Protein for Potential Treatment of COVID-19"

_pharmaceuticals, 2023, doi:10.3390/ph16020296_

Round 1

Reviewer 1 Report

In the submitted manuscript entitled “In Silico screening of drugs that target different forms of E protein for potential treatment of COVID-19”, authors have predicted multitarget ligand against E protein of SARS-CoV-2.

 They have talked about coronavirus (CoV), structure and function of envelope protein (E) and uses of existing and already probed drugs against E protein of SARS-CoV-2.

Methodology of the article consist structure prediction of E protein of SARS-CoV-2, ADMET analysis, Molecular docking and MD simulations are done.

 There are few important questions needed to be answered –

11)      What is use of ADMET analysis on FDA approved drugs? Also for world-not-FDA drugs. A molecule becomes approved drug after pre-clinical and clinical trials. ADMET prediction based screening should be explained.

22)      Sec 4.2 says remaining 1921 compounds are considered for next step. Sec 4.3 starts with “Molecular docking studies of the 2155 ligands were carried”. Write the correct numbers in complete manuscript.

33)      Also sec 4.3 reports, Amantadine as a control for E protein of SARS-CoV-2 in docking studies. Authors should consider this molecule as a control to predict better inhibitors for selected protein and compare their proposed molecules.

44)      What are the selection criteria for those 4-5 ligands.

55)    Modelled proteins are simulated. Docking analysis must be done with after simulations of three E protein MD simulations (without a ligand).

66)     Simulation details like number of lipid molecules, total size of system and any structural changes in protein after simulation should be reported.

77)      Table 2 is giving the MMPBSA for the E-Homopentamer. Receptor remains same in first six docked complexes but the energy values for receptor also changing from 7249 to 7511 (kcal/mol). What is the reason behind these values.

Minor points

aa)      Introduction section – “The subfamily Coronavirinae are divided in three genera” – it should be four

bb)      2.3.4 - (C-terminal harpin) – it should be hairpin. Similarly at several other places

cc)      Relevant refs like 10.3390/covid2020011, 10.1016/j.medidd.2022.100146 may be added in reference section

Reviewer 2 Report

The manuscript entitled “In Silico screening of drugs that target different forms of E pro- tein for potential treatment of COVID-19” by Ramirez-Salinas and Colleagues presents an interesting research on the E protein of SARS-CoV-2 as target in the potential treatment of COVID-19. Authors investigated the two forms of the E protein using an in silico approach and, in particular, they used 3D structure of E protein for a virtual screening. They identified some chemicals as promising binders.

The manuscript in of interest for the scientific community but I have some major issue: 

General:

- Replace In Silico with in silico;

- Verify the number of significant digits displayed in floating numbers, e.g. replace 91.379 % with 91.4 %; 

- A figure representing the pipeline with goals could help the reader.

- Authors should improve/check the quality of supplementary materials (I see all the table unformatted);

- There are too many typos. Authors should massively review their text.

Results section:

- 2.1: Authors should verify the geometrical and energetical parameters to prove the robustness of their models; Authors should add in supplementary materials the alignment between query and template;

-2.4: Authors should compute, for each chemical, the conserved interaction during MD simulations. Moreover, some MD simulations seems not equilibrated from RMSD plot. Authors should extend the simulations up to 300 ns (please verify that are enough, otherwise MD should be extended) and/or perform some (minimum 3) replicas.

Reviewer 3 Report

Dear authors,

The article " In Silico screening of drugs that target different forms of E protein for potential treatment of COVID-19” investigates potential inhibitors of SARS-CoV-2 E protein as a possible treatment for COVID-19. The article is helpful, with well-described methods and encouraging results. 

However, I have some comments for the article that should be included before it is published:

  1. Please write the names of the compounds in lowercase

Abstract:

  1. The abstract section is unclear, and I suggest that the abstract be rewritten. 

Introduction:

  1. This paragraph is too long: “Since then, few endemic CoV have emerged in the last two decades, the first one, the denominated SARSCoV in 2002 with about 8000 cases all over the world and with a mortality rate of 9.6%, and lately 2012 the MERS-CoV emerged, which infected about 2500 over the world and with a mortality rate of 40%, whereas the current pandemic of SARS-CoV-2 shows a mortality of 6.9% [2] which represents the lowest rate in comparison to other coronaviruses but with the highest efficiency of virus spread infecting 650, 941, 156 people and 6,649,897 of deaths (December 7th2022) [3, 4], leading to an important social, economic and health problem all over the globe.” please rephrase

Overall, I'd like to appreciate the authors for their efforts. The work is interesting, and the findings appear to be promising enough to serve as a starting point for further research.

Best regards,

The reviewer

Round 2

Reviewer 1 Report

Authors have responded all the queries and manuscript can be accepted

Author Response

Thank you for your comments

Reviewer 2 Report

In this second round, I have only some minor issues:

1) All the figures should be re-formatted since are out of margin;

2) Table 2, Table 3 and Discussion: please consider the number of significative digits according to MMPBSA precision (doi: 10.1517/17460441.2015.1032936) (e.g. -7885.92 -> -7885; -15.1966 -> -15);

3) deltaG should be replaced with ΔG;

4) Figure S1, since it is a pipeline, I think that some arrows should be inserted to guide the reader. I also suggest to use a "funnel" to give the idea of chemical selection ( like Figure 7 of https://journals.plos.org/plosone/article?id=10.1371/journal.pone.0104822, but obviously adapted to Authors pipeline).
